

# Biodegradation of crystal violet by newly isolated bacteria

Soon Jun Kwak[1], Jiyul Park[2], Yuri Sim[3], Hisu Choi[4], Jein Cho[5] and Young-Man Lee[6]

[1] Department of Clinical Research Design & Evaluation, Samsung Advanced Institute for Health Sciences & Technology (SAIHST), Sungkyunkwan University, Seoul, Republic of South Korea
[2] Dongwoo Girl's High School, Suwon, Gyeonggi-do, Republic of South Korea
[3] Hwasung High School, Hwasung, Gyeonggi-do, Republic of South Korea
[4] Gunpo High School, Gunpo, Gyeonggi-do, Republic of South Korea
[5] Hyowon High School, Suwon, Gyeonggi-do, Republic of South Korea
[6] Dasan University College, Ajou University, Suwon, Gyeonggi-do, Republic of South Korea

Corresponding author
Young-Man Lee, lymbest@ajou.ac.kr

## ABSTRACT

Confronting the environmental threat posed by textile dyes, this study highlights bioremediation as a pivotal solution to mitigate the impacts of Crystal Violet, a widely-utilized triphenylmethane dye known for its mutagenic and mitotic toxicity. We isolated and identified several bacterial strains capable of degrading Crystal Violet under various environmental conditions. Newly identified strains, including *Mycolicibacterium nivoides*, *Chryseobacterium* sp., *Agrobacterium rhizogenes*, *Pseudomonas crudilactis*, and *Pseudomonas koreensis* demonstrated significant decolorization activity of Crystal Violet, complementing the already known capabilities of *Stenotrophomonas maltophilia*. Initial experiments using crude extracts confirmed their degradation potential, followed by detailed studies that investigated the impact of different pH levels and temperatures on some strains' degradation efficiency. Depending on the bacteria, the degree of activity change according to pH and temperature was different. At 37 °C, *Chryseobacterium* sp. and *Stenotrophomonas maltophilia* exhibited higher degradation activity compared to 25 °C, while *Pseudomonas crudilactis* and *Mycolicibacterium nivoides* did not exhibit a statistically significant difference between the two temperatures. *Mycolicibacterium nivoides* performed optimally at pH 8, while *Pseudomonas crudilactis* showed high activity at pH 5. *Stenotrophomonas maltophilia*'s activity remained consistent across the pH range. These findings not only underscore the effectiveness of these bacteria as agents for Crystal Violet degradation but also pave the way for their application in large-scale bioremediation processes for the treatment of textile effluents, marking them as vital to environmental sustainability efforts.

## INTRODUCTION

Crystal Violet, a triphenylmethane dye, is widely used in histological staining and bacterial classification, notably in Gram staining procedures. Despite its widespread use in medical and laboratory contexts, Crystal Violet has been identified as having significant toxic

effects, making its environmental and health impact a subject of concern (*Au et al., 1978*; *Bumpus & Brock, 1988*; *Mani & Bharagava, 2016*). Known for its stability and vivid coloration, Crystal Violet poses a dual challenge as both a potential carcinogen and an environmental contaminant, persisting in water and soil where it can adversely affect aquatic life and disrupt ecosystems. It is documented as hazardous according to Material Safety Data Sheets, and its consumption is strictly due to its toxic nature, as noted by entities like the FDA in the United States and Canadian health authorities.

In medical and antimicrobial solutions, it functions as a mutagen and bacteriostat, helping to prevent fungal growth, and is also employed as a cell-destroying disinfectant (*Au et al., 1978*; *Bumpus & Brock, 1988*; *Mani & Bharagava, 2016*).

Traditional physical-chemical dye removal techniques such as adsorption, chemical coagulation, and advanced oxidation processes, while common, are often costly and inefficient, highlighting the need for more effective and environmentally sustainable alternatives.

The environmental persistence of Crystal Violet, combined with its toxicological profile, necessitates a better understanding of potential bioremediation techniques. In this regard, the study of bacteria capable of degrading Crystal Violet has become increasingly important. The use of Crystal Violet in bacterial culturing serves not only as a method for the identification of bacteria but also as a means to assess their capacity to decompose this dye (*Anderson, 2013*). This is crucial since bacteria that can break down Crystal Violet could potentially be used to counteract its harmful environmental and biological effects.

Recently, biological approaches, particularly cocultivation techniques involving both fungal and bacterial species, have shown promise. These techniques utilize lignin-degrading enzymes such as laccase (Lac), manganese peroxidase (MnP), and lignin peroxidase (LiP), which are more effective at breaking down complex dye structures and accelerating their biodegradation (*Tian et al., 2024*).

Previously, strains of bacteria such as *Enterobacter* sp. (*Jeong et al., 1998*; *Roy et al., 2018*), *Bacillus* sp. (*Yatome, Ogawa & Matsui, 1991*; *Yang et al., 2020*), *Nocardia corallina* (*Yatome et al., 1993*), *Pseudomonas putida* (*Chen et al., 2007*) have been identified as capable of degrading Crystal Violet. Particularly, strains like *Pseudomonas aeruginosa* have shown the capability to utilize enzymes such as azoreductases, which break down the azo bonds (-N=N-) in CV, leading to its decolorization and breakdown into less harmful substances (*Chen et al., 2007*). This biodegradation not only removes the color but also significantly reduces the dye's toxicity, presenting a sustainable solution to CV pollution (*Anderson, 2013*; *Jeong et al., 1998*; *Roy et al., 2018*). However, there exists a knowledge gap in understanding the complete pathways and mechanisms through which these bacteria degrade CV.

Our research builds on these foundations, aiming to not only validate the dye-degrading capabilities of known bacterial strains but also to explore and quantitatively assess the abilities of other, as yet unrecognized, bacterial species in Crystal Violet decomposition. This investigation seeks to broaden the spectrum of Crystal Violet degrading bacteria. By doing so, our study contributes to the growing field of microbial bioremediation, offering new insights into environmentally sustainable methods for managing and

mitigating the risks posed by toxic dyes like Crystal Violet. The broader objective is to enhance our understanding of how these bacteria can be effectively utilized in real-world applications to address the environmental challenges posed by persistent and toxic dyes.

## MATERIALS AND METHODS

### Bacterial culture and selection

To identify bacteria capable of degrading Crystal Violet, a total of nine soil and river water samples were collected from Seoho Stream in Suwon and the wastewater treatment facility of Dongwon F&B's Suwon plant. These samples were then spread onto media prepared with R2A powder mixed with Crystal Violet dye (100 mg/l). The cultures were incubated at room temperature for 6 days. Upon inspection, seven strains that formed single colonies were selected for further experimentation.

### Bacterial cell lysis for assessment of enzymatic activity from crude extracts

To evaluate the enzymatic activity using crude extracts, the cells were first suspended in a 50 mM phosphate buffer solution (pH 7.4) containing 1 mg/ml lysozyme. This suspension was then incubated in water bath for a 37 °C for approximately 30 min. Subsequently, the cells were lysed using a manual homogenizer and sonicated in ice, using a cycle of 5 s on and 10 s off for a total duration of 5 min.

### Bacterial identification

Bacterial species were identified by outsourcing the sequencing to Bionics Co., Korea. Cultured bacterial strains from agar plates were sent to Bionics. We analyzed the 16S rRNA nucleotide sequences obtained from Bionics using the NCBI BLASTn tool (https://blast.ncbi.nlm.nih.gov/Blast.cgi). For the BLASTn analysis, default parameters were used, and the species determination was based on the highest similarity score.

### Protein concentration measurement

To accurately determine the protein concentration in the crude extracts, a standard curve was established using albumin solutions at known concentrations (10, 5, 2.5, 1.25 and 0 mg/ml). For the standard, 50 μl of each solution was injected into 900 μl of Biuret reagent. The absorbance of these solutions was then measured at 545 nm, and the absorbance values obtained from these standards were used to plot a standard curve. By comparing the absorbance readings of the crude extracts against this curve, the protein concentrations within the extracts were precisely calculated.

### Assay of dye degradation/decolorization using crude extracts

To ascertain the capability of bacterial strains to degrade Crystal Violet, crude extracts were first obtained by lysing the cells that were initially isolated on media containing Crystal Violet. These extracts were then quantitatively assayed for their protein content, and a consistent amount of 300 micrograms of protein was used per reaction to evaluate decolorization activity. The extracts were mixed with Crystal Violet dye solutions and incubated at room temperature for 48 h. The rate of decolorization was quantified by the

decrease in absorbance at the dye's maximum absorption wavelength of 590 nm, with the decolorization percentage calculated using the formula (*Chen et al., 2003*):

$$\text{Dye Degradation rate } (\%) = \frac{\textit{Initial OD} - \textit{Final OD}}{\textit{Initial OD}} \times 100.$$

### Evaluating the impact of pH or temperature on the degradation of Crystal Violet

The influence of various pH levels on the degradation of Crystal Violet by the isolated bacteria was investigated by cultivating it in MS medium containing Crystal Violet. Observations were conducted over a period of 5 days at a constant temperature of 25 °C, employing 1.5 ml of MS medium supplemented with a 10% (v/v) inoculum from an overnight culture and 50 mg/l of Crystal Violet dye across a pH range of 5.0 to 8.0. Furthermore, to ascertain the most favorable temperature for degradation activity, assays were carried out at temperatures at 25 °C and 37 °C, under the same initial conditions.

Following the specified incubation period for each condition, the reaction mixtures were centrifuged at 10,000 rpm for 10 min to pellet the biomass, and the supernatant's absorbance at 590 nm was measured. The decolorization percentage was also calculated using the above formula.

### Statistical analysis

A t-test was used to evaluate differences in degradation activity between two temperature settings, 25 °C and 37 °C, to determine if the differences were statistically significant. For the analysis of degradation activity across multiple pH levels, an analysis of variance (ANOVA) was employed to assess whether there were significant differences in bacterial performance across the range of tested pH conditions.

## RESULTS AND DISCUSSION

To isolate bacteria capable of degrading Crystal Violet, samples from soil and river water were cultured on agar supplemented with Crystal Violet (Fig. 1). Bacteria lacking the ability to degrade Crystal Violet are either unable to grow or their growth is inhibited by the dye (*Sharma et al., 2004*). From each plate, approximately 3 to 10 distinct colony types were observed, suggesting a potential for Crystal Violet degradation.

From the colonies that grew on the Crystal Violet-supplemented agar, seven were arbitrarily selected and designated with temporary names such as cvbr, scd1, *etc*. These strains were then purely isolated and subjected to further testing.

To confirm the ability of these strains to decompose Crystal Violet, the bacterial cells of the harvested strains were lysed to obtain a crude extracts, which was mixed with the Crystal Violet solution (*Jeong et al., 1998*). It was observed that the color of Crystal Violet became lighter as time passed in the solution of Crystal Violet mixed with crude extract for all seven pure isolated bacteria (Fig. 2). A control solution containing only Crystal Violet was also prepared to serve as a negative control, where no change in absorbance was observed over time, confirming the stability of the dye in the absence of bacterial activity. As a result, it was confirmed that the bacteria grown in the Crystal Violet medium could

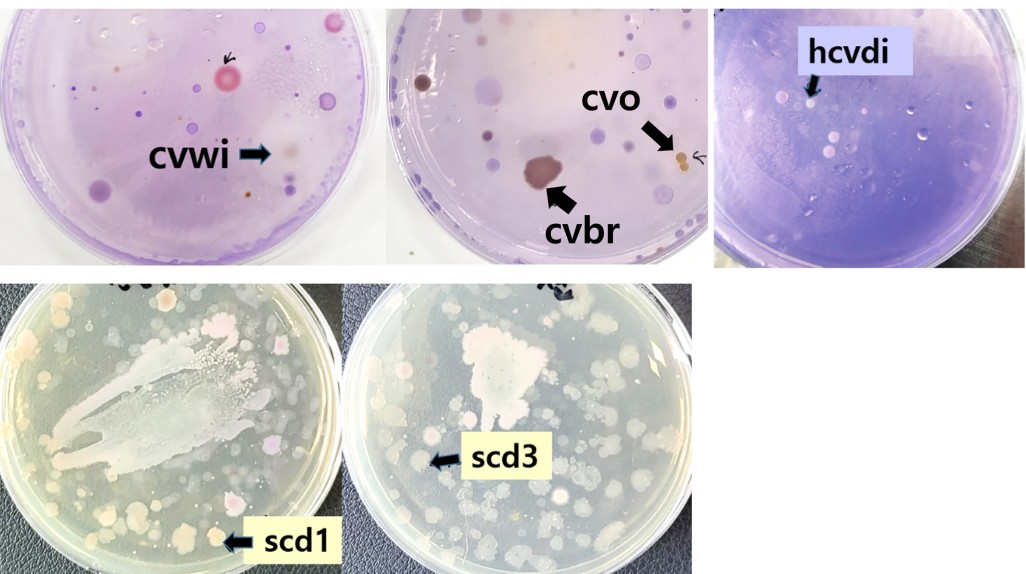

**Figure 1 Bacteria grown on Crystal Violet medium from soil and river water.** Soil and river water samples were diluted and inoculated into agar solid medium containing Crystal Violet and left at room temperature for 6 days. Colonies used in subsequent studies after naming were indicated by arrows.

actually degrade the Crystal Violet, and the identification of the species of these bacteria was attempted.

Bacterial species identified through 16S rRNA nucleotide sequence and BLAST are summarized in Table 1. Among these bacteria, the bacterium named kw was identified as *Stenotrophomonas maltophilia*, and this bacterium was already known to degrade Crystal Violet (*Kim et al., 2002*).

Except for *Stenotrophomonas maltophilia* (kw), the remaining five bacteria were identified as *Mycolicibacterium nivoides, Chryseobacterium* sp., *Agrobacterium rhizogenes, Pseudomonas crudilactis*, and *Pseudomonas koreensis*, and it is not yet known that these bacteria degrade Crystal Violet.

Since the characteristics of these bacteria related to Crystal Violet decomposition have not been identified, the Crystal Violet decomposition ability at various temperatures and pH was examined to discover relevant traits.

Initially, bacterial strains displaying notable Crystal Violet degradation activity (Fig. 2) were selected for further examination of temperature dependence, along with a control *Stenotrophomonas maltophilia* (kw) strain. These were incubated with the dye at both 37 °C and 25 °C to investigate the temperature dependence of their activity. The color change was observed after 5 days. At 37 °C, *Chryseobacterium* sp. (cvbr) exhibited a statistically increase in degradation activity for Crystal Violet compared to 25 °C, as did *Stenotrophomonas maltophilia* (kw), suggesting a notable temperature effect on their enzymatic efficiency. In contrast, *Pseudomonas crudilactis* (cvwi) and *Mycolicibacterium nivoides* (hcvd1) did not exhibit a statistically significant difference in degradation activity between the two temperature, although a slight preference for 25 °C was observed (Fig. 3).

A.

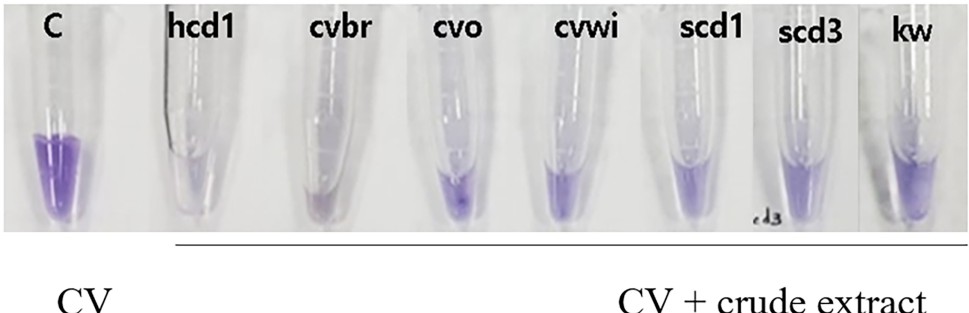

CV                          CV + crude extract

B.

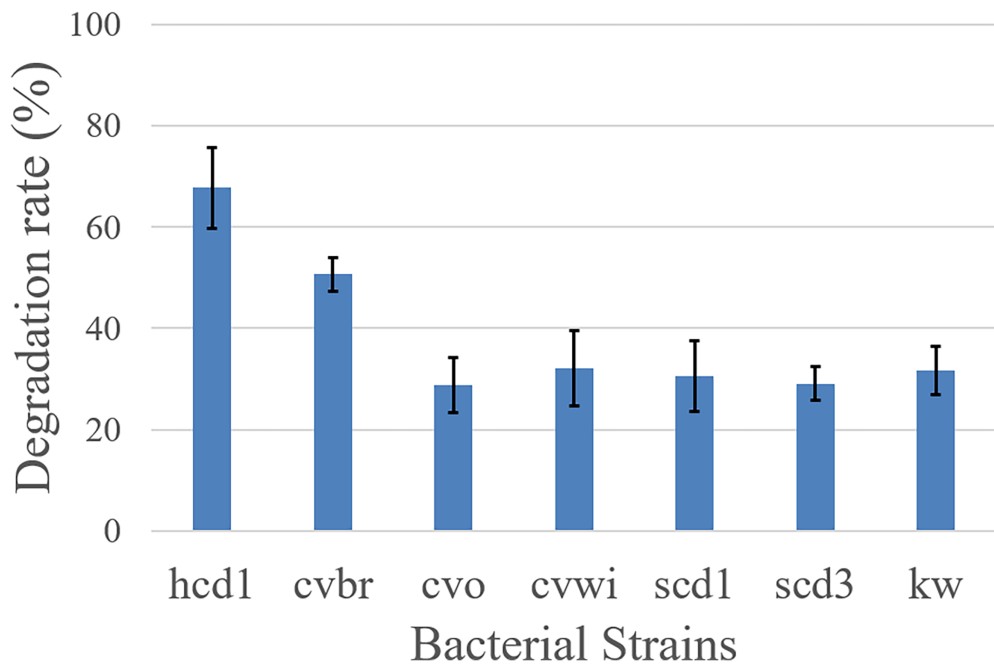

**Figure 2 Confirmation of Crystal Violet (CV) degradation activity of crude bacterial extracts mixed with Crystal Violet.** (A) The color change was observed 2 days at room temperature after mixing the Crystal Violet solution (50 mg/l) with the crude bacterial extract obtained by harvesting each of the bacteria grown in the Crystal Violet-added medium. (B) Quantitative analysis of Crystal Violet degradation by crude bacterial extracts in triplicates representing three separate bacterial cell batches. The bar graph represents the degradation rate (%) of Crystal Violet, indicating the percentage of dye degradation by each bacterial extract. The results are expressed as the mean ± standard deviation(SD). A control consisting solely of Crystal Violet solution showed no absorbance change.

This indicates their potential for stable performance across diverse temperature conditions. These results imply that while temperature is a crucial factor for the degradation capacity of certain bacterial strains, its influence may vary across different

**Table 1 Identification of Crystal Violet degrading bacteria.**

| Name | Species identified | Accession number | Similarity |
|------|---------------------|------------------|------------|
| hcvd1 | *Mycolicibacterium nivoides* | CP034072.1 | 98% |
| cvbr | *Chryseobacterium* sp. | JQ977154.1 | 97% |
| cvo | *Agrobacterium rhizogenes* | MN712287.1 | 98% |
| cvwi | *Pseudomonas crudilactis* | MH016575.2 | 98% |
| scd1, scd3 | *Pseudomonas koreensis* | MH144336.1 | 99% |
| kw | *Stenotrophomonas maltophilia* | CP049956.1 | 98% |

species. This study serves as a foundational investigation for utilizing bacteria in addressing environmental issues, indicating the potential to optimize biodegradation processes under specific environmental conditions through temperature management. For example, *Chryseobacterium* sp., *Stenotrophomonas maltophilia* exhibiting higher degradation activity at higher temperatures could be more effectively utilized in warmer environmental conditions.

In addition, the four bacterial strains previously analyzed for temperature-dependent degradation activity were further subjected to Crystal Violet degradation assays across a range of pH conditions pHs (pH 5, 6, 7, 8) (Fig. 4). A control consisting solely Crystal Violet solution indicated no color change from pH 5 to 8. These data are provided in the Supplementary Data.

The statistical analysis revealed distinct responses to pH changes among the bacteria. For *Stenotrophomonas maltophilia* (kw), the activity remained fairly consistent across the pH range tested, with an F-value of 0.370 and a *p*-value of 0.776, suggesting no significant impact of pH on its degradation capability. *Mycolicibacterium nivoides* (hcvd1) showed a minor decrease in activity at pH 6 and a slight increase at pH 8, with an F-value of 4.604 and a *p*-value of 0.037, indicating a mild sensitivity to pH variations. *Chryseobacterium* sp. (cvbr) showed reduced activity at pH levels below 6, with an F-value of 10.488 but demonstrated a statistically significant increase in activity at higher pH levels, as indicated by a *p*-value of 0.0038. *Pseudomonas crudilactis* (cvwi) demonstrated the highest activity at pH 5, with its degradation capacity being statistically significant, with an F-value of 17.567 and a *p*-value of 0.0007.

These findings not only illustrate the nuanced impact of pH on bacterial degradation activity but also hint at the potential environmental applications. Specifically, bacteria like *Pseudomonas crudilactis*, showing high degradation activity in more acidic conditions, could be considered for bioremediation in environments where acidic pH is prevalent. Conversely, *Chryseobacterium* sp.'s activity in more alkaline conditions suggests it could be suited for different bioremediation applications where higher pH levels are encountered.

The distinct responses to temperature and pH changes not only underline the metabolic adaptability of these organisms but also suggest potential applications. These insights can guide future research into the environmental and industrial deployment of these bacteria, potentially aiding in pollutant degradation and offering a sustainable alternative to chemical treatments.

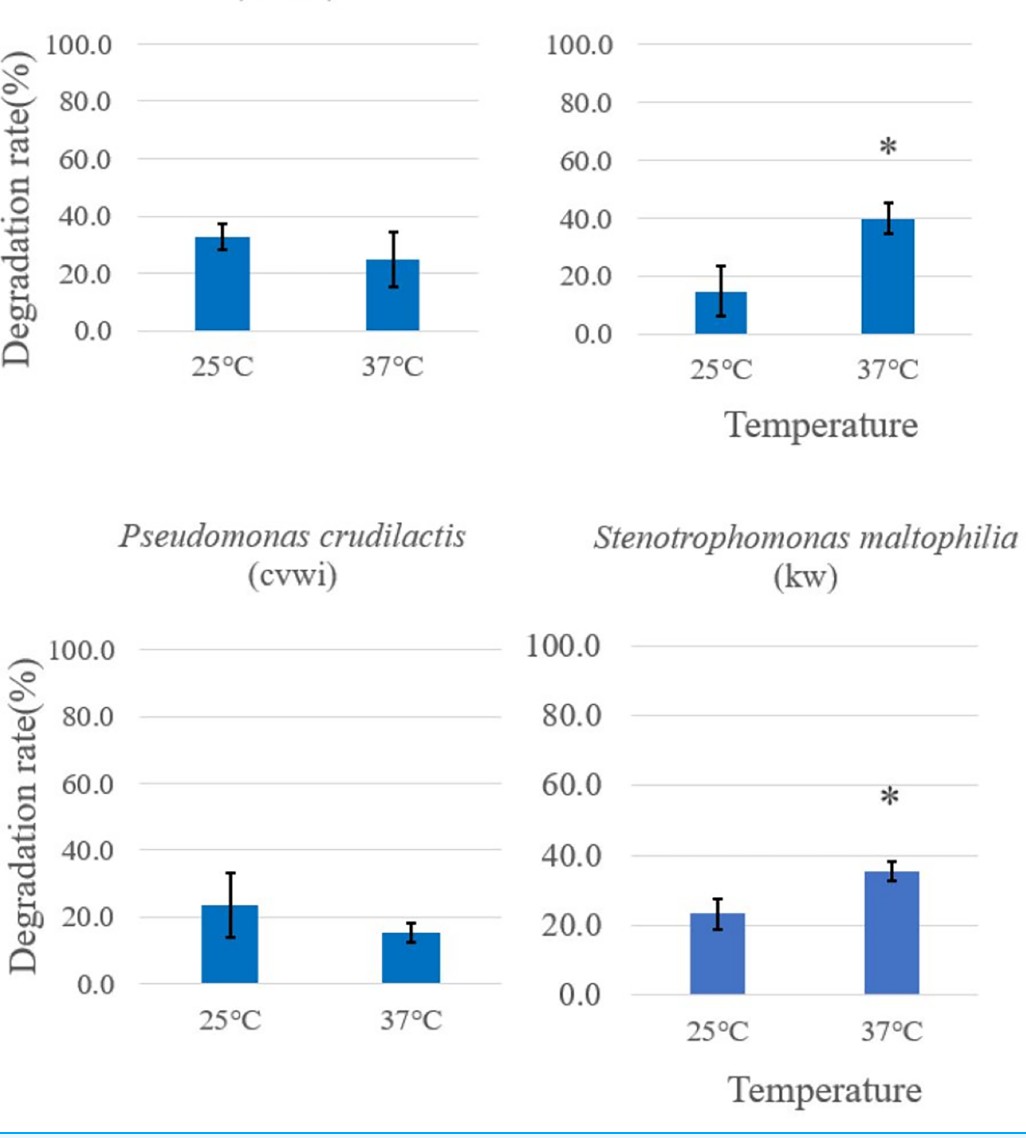

**Figure 3 Temperature-dependent Crystal Violet degradation activity comparison among different Crystal Violet-degrading bacteria.** Crystal Violet solution and bacterial cells grown in Crystal Violet-added were mixed and placed at 37 °C and 25 °C, respectively, and color change was observed after 5 days. The degradation rate was expressed as {"(Crystal Violet absorbance before reaction-Crystal Violet absorbance after reaction)/Crystal Violet absorbance before reaction" × 100%} by measuring the absorbance of crystal violet in triplicates. Each measurement was conducted in three separate bacterial cell batches. The results are expressed as the mean ± standard deviation (SD). An asterisk was placed above the bars in the graphs when the *p*-value was below 0.05. *p*-values for the bacterial strains are as follows: *Mycolicibacterium nivoides*-0.3634, *Chryseobacterium* sp.-0.0344, *Pseudomonas crudilactis*-0.3402, *Stenotrophomonas maltophilia*-0.0416. A control consisting solely of Crystal Violet solution showed no absorbance change.          

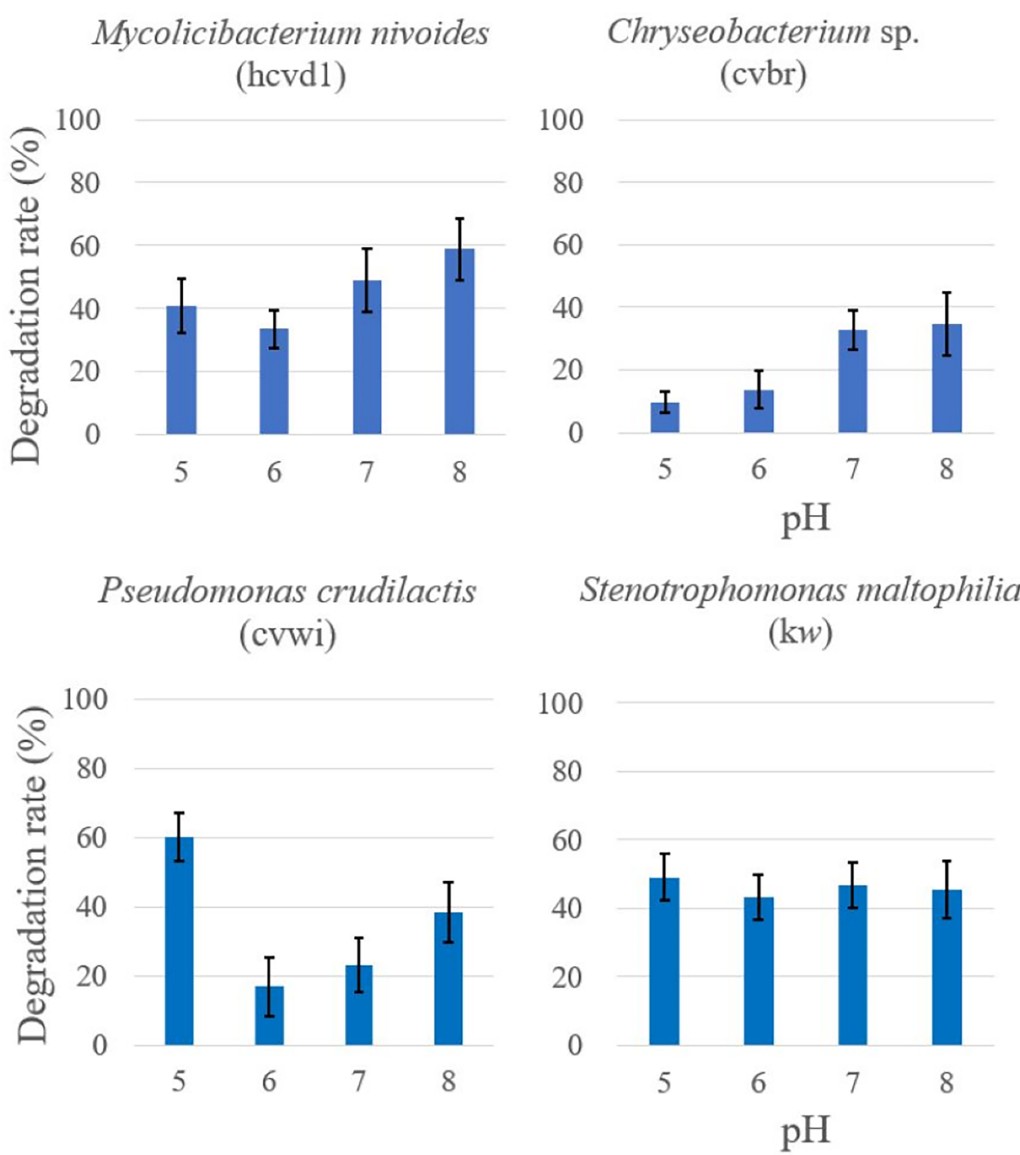

**Figure 4 Comparison of crystal violet degradation activity as a function of pH.** Each bacterium and Crystal Violet solutions of various pHs were mixed and reacted for 5 days, and then the color change was compared. The degradation rate was expressed as {"(Crystal Violet absorbance before reaction-Crystal Violet absorbance after reaction)/Crystal Violet absorbance before reaction" × 100%} by measuring the absorbance of crystal violet in triplicates. Each measurement was conducted in three separate bacterial cell batches. The results are expressed as the mean ± standard deviation (SD). A control consisting solely of Crystal Violet solution showed no absorbance change.

## CONCLUSIONS

In this study, we successfully identified bacterial strains capable of degrading Crystal Violet under various environmental conditions. These include *Mycolicibacterium nivoides*, *Chryseobacterium* sp., *Agrobacterium rhizogenes*, *Pseudomonas crudilactis*, *Pseudomonas koreensis*, and *Stenotrophomonas maltophilia*.

We confirmed the Crystal Violet degradation activity of crude extracts from these strains.

Further investigation demonstrates distinct activities across different temperatures and pH levels. *Chryseobacterium* sp. and *Stenotrophomonas maltophilia* have higher degradation activities at 37 °C, suggesting their suitability for warmer environments. Meanwhile, other strains, such *as Pseudomonas crudilactis* and *Mycolicibacterium nivoides*, showed no significant differences in activity between temperatures, but displayed a slight preference for cooler conditions.

Analysis of pH influences revealed various responses among the strains, further underscoring the complexity of their enzymatic activities. Notably, *Pseudomonas crudilactis* was most active at an acidic pH of 5, whereas *Chryseobacterium* sp. showed increased activity in more alkaline conditions, indicating a potential for tailored application in pH-specific environments. Our findings provide a foundation for further exploration into the use of these bacteria for environmental bioremediation, potentially serving as a sustainable method for pollutant degradation.

In conclusion, this study has uncovered novel Crystal Violet degrading bacterial strains, opening up opportunities for their practical utilization. Future research endeavors may focus on harnessing these bacterial strains for environmental pollutant removal, inhibition of biofilm formation, or other potential applications in industrial and medical fields. Such efforts are expected to contribute to environmental protection and the development of practical application programs.

### Funding

The authors received no funding for this work.

### Competing Interests

The authors declare that they have no competing interests.

### Author Contributions

- Soon Jun Kwak conceived and designed the experiments, performed the experiments, analyzed the data, prepared figures and/or tables, authored or reviewed drafts of the article, and approved the final draft.
- Jiyul Park performed the experiments, authored or reviewed drafts of the article, and approved the final draft.
- Yuri Sim performed the experiments, authored or reviewed drafts of the article, and approved the final draft.
- Hisu Choi performed the experiments, authored or reviewed drafts of the article, and approved the final draft.
- Jein Cho performed the experiments, authored or reviewed drafts of the article, and approved the final draft.

- Young-Man Lee conceived and designed the experiments, performed the experiments, analyzed the data, prepared figures and/or tables, authored or reviewed drafts of the article, and approved the final draft.

## Data Availability

The raw measurements are available in the Supplemental Files.

## Supplemental Information

Supplemental information for this article can be found online at http://dx.doi.org/10.7717/peerj.17442#supplemental-information.

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
