# Peer review of "Biodegradation of crystal violet by newly isolated bacteria"

_PeerJ, doi:10.7717/peerj.17442_

## Round 0.1 · original submission · Major Revisions

Both reviewers were positive in terms of the study filling a gap in current scientific knowledge. However, both had numerous concerns on a variety of points that will need addressing before the manuscript is in a sufficient condition to accept.

Reviewer 1 ·

Basic reporting

The manuscript “Biodegradation of crystal violet by newly isolated bacteria” by Kwak et al., describes the ability of six environmental bacterial species to degrade the toxic dye crystal violet. They have discussed this with clear, well-written language that clearly conveys their ideas. The authors have provided some background to the ecological impact of crystal violet and have framed their study with relevant context. I would recommend further detail is included on the degradation mechanism of crystal violet, especially due to the number of bacterial species discussed in lines 58 and 59. The authors have provided raw data within supplementary tables; however, these are unclear in their labelling (especially concerning supplementary figure 3) and with some cell titles remaining in Korean which will need to be translated for clarity. All provided material could be accessed in a readable format. This manuscript matches the desired format of PeerJ; however, funding support may be contained with the Acknowledgements which goes against the guidance. Typing errors can be found in lines 182 and 184. Bacterial names remain full-length throughout the manuscript, this will need to be changed to fit current nomenclature.

Experimental design

The study carried out by Kwak et al. fits the scope of PeerJ, with the crude enzymatic extracts having potential applications in industry in the bioremediation of eco-toxins such as crystal violet. Whilst the authors describe some novel bacterial species that possess proteins capable of degrading crystal violet, the techniques and results described are not wholly novel. A similar study was published in PeerJ in 2018 (https://www.ncbi.nlm.nih.gov/pmc/articles/PMC6015751/), with more extensive testing of environmental conditions. Whilst the authors have used triplicate data to produce graphs, there is no mention of whether these are biological or technical repeats. Certain negative controls seem to be absent from Figures 2b, 3 and 4. The authors need to expand the level of detail for their methods – a concentration of lysozyme is required for line 82 and parameter descriptions are needed for the NCBI BLASTn described in line 89. The error bars present in Figures 2b, 3 and 4 need to be darker in colour as they cannot be clearly visualised.

Validity of the findings

The authors have described several new species of bacteria associated with the ability to degrade crystal violet, that span both Gram-positive and Gram-negative lineages. Of the six bacteria described in this study, only M. nivoides hcdv1 and Chryseobacterium sp. cvbr are novel to the field. The remaining four genera have been implicated in crystal violet degradation previously. The authors have tested the crude enzymatic extracts against crystal violet over two temperatures and four pH conditions. The authors make several claims about the efficacy of the enzyme activity despite no statistical analysis being included in the manuscript. Therefore the incorrect statements used in lines 151, 155 and 163 need revised as these statements are invalid. As previously mentioned no negative controls have been plotted for Figure 4, despite the authors alluding to its use in the main text. This figure will need revision, especially since at pH 9 crystal violet self-decomposes which will affect the interpretation of the results.

Additional comments

Major revisions:
• Figure 1 is difficult to interpret due to the manner in which the photograph was taken. Improved labelling of colonies by electronic means may make this clearer and the authors may wish to take a high-resolution photo to observe colony morphology characteristics.
• For Figure 2a, a negative control has been used in the 48-hour observation of degradation – this has not been plotted in Figure 2b. Figure 2b lacks any form of legend/key which makes interpretation difficult. Authors may consider relabelling charts to include bacterial names as opposed to code which is confusing. Error bars need to be made clearer and statistical tests would allow for accurate comparisons between species.
• Similar controls are missing from Figure 3 as well as several bacteria (cvo, scd1 and scd3) – this needs addressed in the manuscript. Authors may consider condensing the graphs so that all bacteria are shown for a single temperature allowing for easier comparisons. Again statistical analysis and error bars need addressed.
• The same strains are absent from Figure 4, again this needs to be addressed in the text.
• Supplementary tables are difficult to understand, cell headings need revision to be made clear and Korean texts need translation into English.
• An additional experiment the authors could consider is a time-based curve allowing for reading to be made over several days. This would allow for an increased understanding of rates of decolourisation as opposed to a single time point. Similarly, these organisms were isolated from a river – a colder temperature would be more relevant to their actual environment and rule out self-degradation of crystal violet as a consequence of increased temperature.

Minor revisions:
• The introduction needs to be expanded to include further details on the degradation mechanisms utilised by bacteria against crystal violet, with a discussion examining pathways.
• Other dyes within the triarylmethane family could be investigated since these are also used in industrial settings. This would allow the authors to draw conclusions on whether enzymatic extracts specifically target crystal violet.
• Authors could treat crude enzymatic extracts, is activity lost after denaturing to confirm protein role in degradation?
• I have concerns regarding the lysozyme treatment of M. nivoides since mycobacteria are known to be resistant to the protein especially since the authors have not reported a working concentration. Similarly, 37oC would not be capable of weakening the cell envelope of mycobacteria. Could the authors make a comment regarding confirmation of effective lysis?

Reviewer 2 ·

Basic reporting

The content of the article was written to a good standard of professional English, however there are areas where this could be improved and where sentences could be written better including the sentence which spans from 108 to 109, and from 126 and 127 (as highlighted in the annotated PDF).

Providing further background information about the field would be preferable to be added in the introduction section which could include, but not limited to, providing some context about the environmental and health impact of Crystal Violet, the current standard process of degrading crystal violet and why improvements are needed in this field/industry (i.e. what problem are you providing a solution for).

The current article structure combines the results with the discussion section. It would be advisable for the authors to split this into two so that the results section appears separate to the discussion section. This is part of the guidance provided by the journal which appears under the ‘Standard Sections’ heading of the ‘Author Instructions’ documentation.

Supplementary data has been appropriately supplied. In the document named ‘peerj-96304-supdata-fig4’, in row 1, you have labelled the results as 4, 2, 1, 7. It would be advisable to include a description in the document as to what bacterial strains 4, 2, 1 and 7 refer to. This is similar to the data in the document with the filename ‘peerj-96304-supdata-fig3’.

For the graph titles appearing in Figure 3 and 4, it would be preferable to use the full name of the bacterial species, instead of including both the acronym and the full name in brackets. For Figure 2, it would be helpful to include a key to showcase which bacterial species correspond with the acronym used. This could either be in the figure itself or the figure legend.

Experimental design

This article contains original research which is covered within the aims and scope of the journal.

The research question has been defined, however the knowledge gap which the research is filling is slightly unclear. This could be improved by providing further context and background information about the field as mentioned in the comments under ‘Basic Reporting’.

A good investigation to answer the research question has been conducted. However, in the introduction, it states that the investigation seeks to determine and understand the degradation mechanisms of the Crystal Violet degrading bacteria. However, it is unclear which experimental methods have been employed to align with this objective.

The methods section was good, however there are parts where specific details would be advisable to include instead of using broad, ambiguous statements or phrases. This in turn could help with reproducibility. These include:

Lines 76-77 - you mention cultures were incubated for ‘about one week’. It would be helpful to, instead, mention the number of days or hours for the incubation period. This would help distinguish between a working week which is 5-days long and a calendar-week which is 7-days long.

Line 81 - Specify how the degrading enzymes were purified from the crude extracts. You’ve mentioned that you were looking to extract the degrading enzymes but there is currently no detail about how these degrading enzymes were purified from the crude extracts, just how the crude extracts were obtained.

- Line 82 - Specify the buffer and lysozyme concentration

- Line 87 - Specify how samples were sent (purified DNA or strain). This would be helpful to distinguish the work that was carried out by Bionics. Currently it is unclear whether the 16S sequencing for example was conducted by Bionics or the authors as it only stats that Bionics conducted the analysis.

- Line 89 - Specify the criteria used for species identification (coverage, identity and E-value). Specify if the first hit was assumed to be the correct species

- Line 96 - specify if linear regression was used to obtain a formula to calculate the concentration of the protein extracts

- Line 100 - specify which broth was used for the cultures

- Line 102 - specify what the innoculum is (i.e. is this an overnight culture?)

- Line 113 - specify wavelength here

Validity of the findings

All data has been provided. It would be nice for statistical analysis to be conducted so that the differences of activity between the two temperatures tested could be compared and reported where a statistical difference in activity was observed.

For Figure 2B, change y-axis to “Dye Degradation” to match the equation in the methods (this measure is not a rate).

Figures 2B, 3 and 4. Specify the number of replicates done in each case (or show individual points instead of bar charts). Standard deviation instead of standard error would also be preferable.

Please do show the negative control values (without extract) for the absorbance at different pHs, at least in the supplementary data.

Overall, the conclusions are linked to the research question.

Annotated reviews are not available for download in order to protect the identity of reviewers who chose to remain anonymous.

---

## Round 0.2 · accepted · Accept

The reviewer was happy that you had addressed the majority of their comments, apart from a couple of minor issues that should be able to be fixed in proofing. Congratulations.

Reviewer 1 ·

Basic reporting

The revised manuscript provided by Kwak et al improves on the previous version and clarifies technical detail on the ability of five previously uncharacterised bacteria in their ability to degrade crystal violet. The introduction now frames the background and clearly states how this work addresses real world issues with crystal violet ecological contamination. Figures and supplementary tables revisions have now been addressed and contain sufficient detail. In the supplementary figure 4 file, the error bars appeared horizontal which may need altered. The manuscript remains in PeerJ format, with no immediate spelling/grammatical errors.

Experimental design

The addition of units and concentration details, alongside detail in experimental procedure, would allow for reproducibility. Inclusion of statistical tests and full raw data in supplementary material allows for comparisons to be made across the conditions, together with the clarification the authors provided for their use of control in their rebuttal statement.

Validity of the findings

The authors have newly identified five bacteria capable of degrading crystal violet across two differing environmental conditions – temperature and pH. The authors confirm that crude extracts from these bacteria degrade at similar rates to a positive control (and in some instances at a higher rate). Whilst no further experiments have been carried out, the authors have clarified previous points with detail that supports the conclusions they have made. Strengthened by statistical tests, the authors shown both conditions impact crystal violet degradation.

Additional comments

Minor comments:
• The two terms ‘triphenylmethane dye’ and ‘triarylmethane dye’ are used in the abstract and line 46, respectively. Whilst these two terms describe the same chemical groups, this may be confusing to some readers not familiar with the literature. Authors should correct so one term is consistently used.
• Crystal violet is not capitalised in line 146, which has been consistently the same throughout manuscript.